# Double-Layered Microcapsules Significantly Improve the Long-Term Effectiveness of Essential Oil

**DOI:** 10.3390/polym12081651

**Published:** 2020-07-24

**Authors:** Ting Zhang, Yu Luo, Mingxing Wang, Feng Chen, Jinkang Liu, Kai Meng, Huijing Zhao

**Affiliations:** 1National Engineering Laboratory for Modern Silk, College of Textile and Clothing Engineering, Soochow University, No. 199 Ren’ai Road, Industrial Park, Suzhou 215123, China; 20175215022@stu.suda.edu.cn (T.Z.); 20185215017@stu.suda.edu.cn (Y.L.); 2Violet Home Textile Technology Co., Ltd., Nantong 201410, China; wangmingxing@violet.com.cn (M.W.); chenfeng@violet.com.cn (F.C.); liujinkang@violet.com.cn (J.L.); 3Nantong Textile & Silk Industrial Technology Research Institute, Nantong 226001, China

**Keywords:** double-layered microcapsules, sustained release, essential oil

## Abstract

In order to improve the long-term effectiveness of essential oil, a double-layered microcapsule was prepared using the inclusion encapsulation method in this study, with β-cyclodextrin as its inner layer and chitosan and sodium alginate as its outer layer. The optimized preparation process was obtained through the response surface method. The morphology, particle size, encapsulation efficiency, thermal stability and sustained release effect of the double-layered microcapsules were characterized. The microcapsules were spherical, with a particle size distribution between 2–6 μm, and had good thermal stability within 250 °C. Their encapsulation efficiency can be up to 80%, and it can continuously release the active ingredients of the essential oil under normal temperature and high temperature conditions for a long time. In order to further examine the application effect of the double-layered microcapsule, it was loaded onto the cotton fabric by the soak-roll method. The finished cotton fabric showed excellent washability and rubbing fastness. They can still maintain a light fragrance naturally for two months. The microcapsules prepared in this study can be potentially applied in sleep aid, antibacterial, mosquito prevention, food science and other related products.

## 1. Introduction 

Plant essential oil is a complex natural mixture. It is colorless and transparent, soluble in organic solvents and highly volatile [1]. As a secondary metabolite produced by aromatic plants, the main components of plant essential oils are terpenes and aromatic compounds [2,3,4]. In the Middle Ages, the Arabs first obtained plant essential oils through steam and water extraction. For thousands of years, because of their aromatic odor, antisepsis, sleeping aid, antibacterial [5], antioxidant, anti-inflammatory, insect repellent and other medicinal properties, plant essential oils have been applied in many fields, such as perfume production, daily chemical industry, antiseptic treatment, food preservation, medical pharmacy, etc. [6,7,8]. In recent years, with the continuous improvement of people’s living standards, plant essential oil, as a safe, natural product, has attracted more and more attention. However, the development and application of plant essential oils are greatly restricted due to their unstable nature, easy decomposition and volatilization [9]. The application of microcapsule technology has opened up an effective way to solve this problem. Microencapsulation technology can protect plant essential oils from the external environment, greatly improve its stability and storage and provide a strong support for the further development and utilization of plant essential oils.

Microencapsulation refers to the use of natural or synthetic polymer materials to encapsulate the target material, which can be gas, liquid droplets or solid particles, into tiny containers with core-shell structures ranging from a few to several hundred to thousands of microns in diameter [10]. At present, there are more than 200 kinds of microcapsules. According to the formation mechanism of microcapsules, the preparation methods can be divided into three categories: physical method, chemical method and physical chemical method [11]. The most common preparation methods are interfacial polymerization, in-situ polymerization, spray drying, inclusion encapsulation, complex coagulation, etc. Each of these methods has advantages and disadvantages. The interaction of core material, preparation method and wall material have different effects on the size of microcapsules, the content of the essential oil (encapsulation efficiency), the preparation process, the cost and the sustained release performance. For example, if β-cyclodextrin is used as the wall material, it has the advantages of low cost, simple process, nontoxic and harmless; meanwhile, the long-term and high-temperature sustained release properties of the microcapsules are not ideal because of the existence of holes in the material [12,13].When it comes to the complex coagulation method, the preparation conditions are mild, the microcapsule encapsulation efficiency is high and it has good controlled release and sustained release effects, because it does not involve high temperatures [8]. However, the disadvantage of the complex coagulation method is that the size of the microcapsules is large. Some toxic crosslinking agents (such as formaldehyde, glutaraldehyde, etc.) are often added during the preparation process in order to improve the mechanical properties and stability of the microcapsules [14,15,16]. The sizes and release behaviors of microcapsules prepared by the existing preparation methods were not satisfactory. The sustained release performance needs to be further strengthened, especially when the microcapsules are applied to some fields that hope to maintain the function of core materials for a long time.

Studies have shown that double-layered microcapsules can effectively improve the sustained release performance of microcapsules and play a better protective effect on essential oils. Fioramonti [17] prepared the whey protein/sodium alginate double-layered microcapsules by a spray-drying method, so as to provide better protection and storage stability for the embedded linseed oil. Chong [18] synthesized double-layered polyurethane/poly (urea-formaldehyde) (PU/PUF) microcapsules containing clove oil by in-situ and interfacial polymerization reactions and found that the release time of clove oil from the double-layered microcapsules was much longer than that of the single-layered ones.

In this study, β-cyclodextrin, chitosan and sodium alginate were selected as the composite wall materials, and a kind of oil microcapsule with a double-layered structure was prepared by using the inclusion of β-cyclodextrin with essential oil, as well as the electrostatic force between chitosan and sodium alginate. At the same time, the morphology, particle size, encapsulation efficiency, thermal stability and sustained release performance of the double-layered microcapsules were characterized. Further, the double-layered microcapsules were finished on fabrics, which broadened the application of microcapsules in sleeping aid, antibacterial, insect repellent, etc.

## 2. Materials and Methods

### 2.1. Materials

Lavender essential oil (LO) was purchased from Shenbao Flavor & Fragrance Ltd., Shanghai, China. β-cyclodextrin (β-CD, CAS number: 7585-39-9) was purchased from Dibai Biotechnology Ltd., Shanghai, China. Chitosan (CS, deacetylation degree 75–85%, CAS number: 9012-76-4) was purchased from Yuanye Biotechnology Ltd., Shanghai, China. Sodium alginate (SA, viscosity 200 ± 20 mPa·s, CAS number: 9005-38-3) was purchased from Meryer Chemical Technology Ltd., Shanghai, China. Acetic acid (CAS number: 64-19-7) was obtained from Shenbo Chemical Ltd., Shanghai, China. Anhydrous ethanol (CAS number: 64-17-5) was obtained from Yonghua Chemical Technology Ltd., Jiangsu, China. Fatty alcohol polyoxyethylene ether (JFC) penetrant was obtained from Kexin Chemical Ltd., Jiangsu, China. Adhesive Goon720 was acquired from Jiahong Technology Ltd., Guangdong, China. Cotton fabric was acquired from Violet Home Textile Technology Ltd., Jiangsu, China.

### 2.2. Preparation of Double-Layered Microcapsules

First, β-cyclodextrin (12 g) was added to deionized water (180 mL); then, they were heated (below 7 °C) and stirred to dissolve. After the solution was clear and cooled down, the lavender essential oil solution of 2.3 mL (dissolved in 10% ethanol aqueous solution) was slowly added in and stirred at the specified temperature for 2–4 h and then refrigerated in a 4 °C refrigerator overnight to precipitate. After vacuum filtration, the obtained precipitate was washed three times with absolute ethanol and then washed once with deionized water and dried in an oven to obtain single-layered microcapsules of lavender essential oil (LOM). Then, chitosan (2 g) was dissolved in a 200-mL 1% acetic acid solution and stirred thoroughly for 2 h to obtain a clear chitosan solution. Then, the above single-layered microcapsule solution was mixed with the chitosan solution, and the mixture was then ultrasonicated for 5 min and oscillated at a constant temperature (37 °C) for 30 min. Then, the mixture was centrifugated at 9000 rpm for 10 min, vacuum-filtered with a Brinell funnel by a vacuum pump (Tan’s Vacuum Equipment Ltd., Linhai, China) and washed with deionized water. After adding the same amount of sodium alginate solution, the steps of oscillation, centrifugation, filtration and washing were repeated as above to obtain the double-layered microcapsule of lavender essential oil (LOM-M).

### 2.3. Response Surface Optimization Experiment

Response surface methodology (RSM) was used to determine the optimal encapsulating conditions of the single/double-layered microcapsules. The effects of three independent variables, LO mass (A), encapsulating time (B) and encapsulating temperature (°C), on the response variable, encapsulation efficiency (EE), were evaluated using the Box-Behnken design (BBD). The Design Expert 8.0 (Stat-Ease, Inc., Minneapolis, MN, USA) was used to analyze the results. The factors and levels are shown in Table 1. 

### 2.4. Characterizations of Microcapsules

#### 2.4.1. Encapsulation Efficiency

The encapsulation efficiency of the microcapsules was quantitatively determined by UV-visible spectrophotometer. The steps are as follows: 

(1) Determination of the standard curve

Lavender essential oil was diluted with absolute ethanol to prepare standard solutions with different concentration gradients. The solutions were put in a cuvette for UV full-spectrum scanning (UV-Vis, Cary 5000, NYSE: A, Santa Clara, CA, USA) with a wavelength range of 200–300 nm. The maximum absorption wavelength was detected, and a standard curve of absorbance versus concentration was obtained; then, the linear correlation coefficient was calculated.

(2) Extraction of the essential oils from the microcapsules

Microcapsule sample (1 g) was added into absolute ethanol (50 mL) and heated at 40 °C in a water bath for 30 min. Then, the solution was ultrasonicated for 1 h and centrifuged. The supernatants of 100 μL, 200 μL and 300 μL were taken and made up to a constant volume of 5 mL with absolute ethanol to obtain a sample of unknown concentration, respectively.

(3) Calculation of the microencapsulation efficiency

The samples with unknown concentrations were measured by UV spectrophotometer (Cary 5000, NYSE: A, Santa Clara, CA, USA), and the absorbance at the maximum absorption wavelength was obtained. Then, the concentration of essential oil was calculated according to the above standard curve. Then the encapsulation efficiency was calculated according to the following formula.
(1)$$\mathrm{EE}\;(\%)\;=\;\frac{W_{1}}{W_{2}}\times 100\%$$
where W_1_ is the quality of essential oil in the microcapsule, and W_2_ is the total amount of essential oil added in the preparation process.

#### 2.4.2. Morphology and Particle Size

A small amount of microcapsule powder was lightly sprayed on the electron microscope platform with conductive glue, and the excess powder on the surface was blown off with a washing ear ball. Then, the sample was sputtered gold for and observed under a scanning electron microscope (SEM, S-4800, Hitachi, Tokyo, Japan). A small amount of microcapsule powder was ultrasonically dispersed in deionized water, and the dispersion liquid was removed with a pipette gun and dropped on a copper mesh and observed under a transmission electron microscope (TEM, Tecnai-G20, FEI, Santa Clara, CA, USA) after drying. The Malvern laser particle size analyzer (MASTERSIZER 2000, Malvern Instruments Ltd., Malvern, UK) was used to determine the size distribution of the microcapsules.

#### 2.4.3. Fourier-Transform Infrared (FT-IR) Spectroscopy

The infrared spectroscopy test of the microcapsules was carried out by the KBr pellet pressing method. The microcapsule powder and KBr crystals were mixed and ground evenly and compressed for sample preparation. Fourier-transform infrared spectrometer (FT-IR, Nicolet 5700, Nicolet, Waltham, MA, USA) was used to measure the transmission infrared spectrum. The scanning range was 4000–500 cm^−1^.

#### 2.4.4. Thermal Analysis

A thermogravimetric analyzer (TG, DIAMOND 5700, PE, Waltham, MA, USA) was used to test the thermal stability of the microcapsules. The sample was heated from 30 to 400 °C at a rate of 20 °C/min under a nitrogen atmosphere.

#### 2.4.5. Evaluation of Sustained Release of Essential Oil Microcapsules

Sustained release performance is an important indicator for evaluating the quality of microcapsules. Considering that microcapsules can be used under high temperature conditions, the release behaviors both under room temperature and high temperature were studied and compared. 

Release behaviors at room temperature. The microcapsules were put in an open environment at 25 °C, and samples (about 1 g) at regular intervals (5 d, 10 d, 20 d, 30 d and 60 d) were taken, and the contents of essential oils in the microcapsules were detected according to the method described in Section 2.4.1. The released amount of essential oil was calculated, and a curve of essential oil contents changing with time was obtained. 

Release behaviors at a high temperature. The microcapsules were put in an oven at 80 °C, and samples at intervals (0, 5, 10, 20 and 30 min) were taken to obtain the curve of essential oil contents changing with time according to the same method described above.

### 2.5. Application of Double-Layered Microcapsules

#### 2.5.1. The Finishing of Microcapsules on Cotton Fabrics

The microcapsule suspension, Goon720 binder and JFC penetrant were mixed with high speed stirring to prepare a finishing solution. The cotton fabric was cut into a size of 10 cm × 10 cm and immersed in the finishing solution for 1 h. Then, a bench rolling mill was used to soak and roll the fabric twice (with a liquid ratio of 100%). Then, the finished fabric was put into the oven for pre-drying (70 °C, 3 min) and drying (120 °C, 3 min) and washed with deionized water and dried at a low temperature.

#### 2.5.2. Characterizations of the Finished Cotton Fabric

The finished fabric was sputtered gold and observed under a desktop electron microscope (TM3030, Hitachi, Tokyo, Japan) at a voltage of 3 kV. Elemental analysis was performed by energy scattering spectroscopy (EDX, TM3030, Hitachi, Tokyo, Japan). Reference standards GB/T 12490-2014 and GB/T 3920-2008, the fabrics finished with microcapsules, were tested for washing resistance and friction resistance, respectively. For the washing resistance test, the fabric (100 mm × 40 mm) was washed for 0, 5, 10, 20 and 30 cycles and naturally dried indoors. One washing cycle lasted for 30 min, clean water was used and the water level was moderate; the temperature of the washing water was 30–40 °C. The wash resistance of the fabric was characterized by calculating the content and loss of essential oil on the fabric after washing. For the friction resistance test, the fabric (50 mm × 200 mm) was rubbed against the friction head with a lining cloth (50 mm × 50 mm) for 0, 10, 20, 30, 50 and 100 times under vertical pressure (9 ± 0.2 N), and the friction resistance of the fabric was characterized by calculating the content of essential oil on the fabric after friction.

#### 2.5.3. Release Behaviors of Essential Oils on Fabrics

The subjective sensory evaluation was used to evaluate the fragrance retention strength of the finished fabric. Five people, aged 20–25 years, were selected to form an experimental group (including 2 males and 3 females, all are graduate students) at regular intervals (new treatment, 1 week, 2 weeks, 4 weeks, 8 weeks and 24 weeks). The members of the group should conduct independent sensory tests on the strength of the fragrance. It was divided into 5 levels: strong scent, medium scent, weak scent, very weak scent and no scent. The evaluation results of each member on fabric fragrance retention were recorded.

The content of essential oils on fabrics under natural storage conditions was determined by the objective instrument method. The finished cotton fabrics at different time intervals (new treatment, 1 week, 2 weeks, 4 weeks, 8 weeks and 24 weeks) were cut into pieces and put into a beaker; a certain amount of absolute ethanol was added into the beaker heating in a 40 °C water bath for 30 min. Then, it was ultrasonicated for 1h and centrifuged to remove the insoluble materials in the solution, and the supernatant was kept, measuring the remaining essential oil according to the method described in Section 2.4.5. 

## 3. Results and Discussion

### 3.1. Determination of the Optimized Parameters

The independent variables were set based on the single factor experiments. The experimental design and results are shown in Appendix A (Appendix A). The encapsulation efficiency (EE) ranged from 69.47% to 80.26%. The analysis of variance showed that the model equations fit well, and the correlation coefficient reached to 99.9%. From the F-value (the larger the F-value, the stronger the effect on the response variable), it can be seen that, in this study, the order of influence on the EE of the microcapsules was the amount of essential oil > encapsulation temperature > encapsulation time > interaction between the amount of essential oil and encapsulation temperature > interaction between the encapsulation temperature and encapsulation time.

The response surface and contour plots (Appendix A of the Appendix A) showed the influence of the essential oil amount (A), encapsulation time (B) and encapsulation temperature (C) on the encapsulation efficiency. As can be seen from Appendix A, the encapsulation temperature remained unchanged, and the encapsulation efficiency showed a trend of increasing at first and then decreasing with the increase of the amount of essential oil and encapsulation time. It indicated that, when the temperature and time were at a certain point, the encapsulation efficiency reached the maximum. At the same time, it can be seen from Appendix A that the contour lines of the amount of essential oil and the encapsulation time are similar to a circle, indicating that their interaction has no significant effect on the encapsulation efficiency, which is consistent with the results of the regression equation. From Appendix A, the effect of temperature on the encapsulation efficiency is similar to that of the amount of the essential oil or encapsulation time. The only difference is that the contour plots of the amount of essential oil and encapsulation temperature are elliptical, indicating that their interactions have significant effects on the encapsulation efficiency. In addition, the interaction between the encapsulation temperature and time shown in Appendix A and the interaction between the amount of essential oil and the encapsulation temperature have similar effects on the encapsulation efficiency.

From the regression model, the optimized parameters were obtained as follows: lavender essential oil of 2.27 mL, encapsulation time of 3.19 h and encapsulation temperature of 52.95 °C. The encapsulation efficiency value was predicted as 80.75% under the above conditions. Three verification experiments were carried out under this condition; 2.3 mL of lavender essential oil, 3.2 h of encapsulation time and 53 °C of encapsulation temperature were selected and the actual measured encapsulation efficiency was 80% ± 1.76%, which was very close to the prediction value, indicating that this model fit the production process well and can obtain the maximum encapsulation efficiency of essential oil under optimized conditions.

### 3.2. Morphology

It can be seen from Figure 1 that the single-layered microcapsules of lavender essential oil appear as irregular parallelograms or diamonds, and their surfaces are relatively smooth. The shapes of the double-layered microcapsules are roughly spherical, with small particles attached to the surfaces. Due to the stickiness of the chitosan and sodium alginate, some of the microcapsules stacked together. It can be clearly seen from the TEM images that there is an outer shell, which is formed by the electrostatic interaction between chitosan and sodium alginate.

### 3.3. Particle Size

The particle size distribution is shown in Figure 2. It can be seen that the diameters of the microcapsules are about 2–5 μm, and there is not much difference between single-layered and double-layered microcapsules, which may be caused by the use of an ultrasound in the double-layered preparation process, and sufficient emulsification reduced the adhesion between the microcapsules. The particle sizes of the double-layered microcapsules are relatively small; a higher specific area is beneficial for the microcapsules to adhere to the fabrics during the finishing process.

### 3.4. FT-IR Spectroscopy

As can be seen in Figure 3, the characteristic absorption bands at 3500–3000 cm^−1^ were due to the stretching vibration of the intramolecular or intermolecular hydroxyl group (-OH). In the polysaccharide of β-cyclodextrin, the absorption peak of the hydroxyl group (-OH) was at 3336 cm^−1^. In lavender essential oil, the absorption peak of the hydroxyl group (-OH) was at 3457 cm^−1^, which was mainly attributed to linalool. When β-cyclodextrin wrapped up lavender essential oil to form a single-layered microcapsule, the characteristic absorption peak of -OH shifted to 3373 cm^−1^. This is because the hydroxyl group in the essential oil stayed outside the cavity of β-cyclodextrin, and the hydrophobic group with less polarity enters the cavity of β-cyclodextrin. The hydroxyl group outside formed an intermolecular hydrogen bond with -OH in β-cyclodextrin, making the -OH in the microcapsule move to a lower wave number direction. Meanwhile, this structure can make the whole inclusion complex system more stable. The strong absorption peak of lavender essential oil appeared at 1740 cm^−1^, which was caused by the stretching vibration of the carbonyl group (C = O), which was attributed to the ester component in essential oil. The absorption peak at 1740 cm^−1^ appeared both in the single-layered and double-layered microcapsules, indicating that the lavender essential oil had been encapsulated in the two types of microcapsules.

Chitosan has a wide absorption peak at 3343 cm^−1^, which is caused by the overlap between the stretching vibration of the hydroxyl group (-OH) and the amino group (-NH_2_). The antisymmetric stretching vibration peak of C-O-C is located at 1158 cm^−1^. The characteristic peak at 2930 cm^−1^ in the infrared spectrum of sodium alginate is due to the asymmetric stretching vibration of C-H in -CH_2_. Comparing the infrared spectra of double-layered and single-layered microcapsules, there are basically no new absorption peaks, indicating that there is no chemical reaction between chitosan and sodium alginate to form new chemical bonds. Instead, they are attracted to each other and coated on the single-layered microcapsule due to static electricity to form a double-layered coating structure.

### 3.5. TG

Due to the possible further process, the microcapsules need to withstand high temperatures without damaging or losing essential oils remarkably. Therefore, the thermal stability of the microcapsules was studied. As can be seen in Figure 4, in general, the quality loss of essential oil microcapsules can be divided into three stages, which are less than 100 °C, between 100–250 °C and more than 250 °C. Before the temperature reached 100 °C, the microcapsules had a mass loss of 7–9%. It was caused by the residual water in the sample and the evaporation of low-boiling components in the essential oil. The second stage was in the temperature range of 100–250 °C, and the mass loss of the microcapsules was relatively smooth; this was caused by the release of high boiling point components in the essential oils. In the last stage, due to the rapid release of essential oil and the decomposition of the wall material, a sharp mass loss occurred. Therefore, the prepared microcapsules can maintain a certain thermal stability within 250 °C, and single-layered and double-layered microcapsules can protect the essential oil from high temperatures to avoid volatilization or degradation. At the same time, comparing the mass loss of single-layered and double-layered microcapsules before the temperature of 250 °C, it is found that the mass loss of double-layered microcapsules was slightly less than that of single-layered ones, indicating that a double-layered wall can improve the thermal stability of microcapsules and play a better role in protecting essential oil from losing at high temperature conditions.

### 3.6. Evaluation of the Sustained Release of Microcapsules

The full spectrum of the UV absorption of lavender essential oil and its standard curve are shown in Appendix A. The absorbance of the sample is measured at the maximum absorption wavelength, and the concentration and content of lavender essential oil can be calculated according to the standard curve.

The release behaviors of microcapsules at room temperature and high temperatures were shown in Figure 5 and Figure 6. Generally, the double-layered microcapsules released less amounts of lavender essential oil than that of single-layered ones. The microcapsules can keep about 70% essential oil after 30 days of releasing either at room temperature or high temperatures. As can be seen in Figure 5, the content of essential oil in the microcapsule gradually decreases with the increase of time. In the first 0-5 days, the decrease rate was the fastest. After five days, it decreased slower. After 30 days, the loss of essential oil in the single-layered and double-layered microcapsules were about 27.88% and 21.03%, respectively, indicating that, after the microencapsulation of essential oil, the sustained release effect was obvious, and the storage stability was satisfying. The situation at high temperature conditions was similar, as can be seen from Figure 6. All the above results showed that the microcapsules prepared in this study had satisfying sustained release effects, and the double-layered microcapsules had better protection effects for the essential oils.

### 3.7. Comparison of Fabric Finishing with Single-Layered and Double-Layered Microcapsules 

The single-layered and double-layered microcapsules were used to finish the cotton fabrics at the same conditions. It can be clearly seen from Figure 7 that the contents of essential oils on the fabrics finished by the double-layered microcapsules were higher than that of the single-layered ones. This reason was that the wall materials of chitosan and sodium alginate in the double-layered microcapsules were stickier and had better adhesion characteristics to the fabrics. Therefore, it is suggested that double-layered microcapsules should be used for further finishing applications.

### 3.8. The Finished Fabrics

Figure 8 was the electron microscope image of cotton fabric finished by the double-layered microcapsules. Most microcapsules adhered to the surface of the fabric, and the particle distribution was relatively uniform. Smaller microcapsules can also be seen on the surfaces of the fibers, indicating that the microcapsules had moderate particle sizes and were suitable for finishing on fabrics. Appendix A shown in Appendix A is the EDS image of fabrics treated with double-layered microcapsules. The uniformity of the distribution of the nitrogen element in the microcapsule wall material and the silicon element in the binder can confirm that the microcapsules are relatively uniformly distributed on the fabric, thereby avoiding the influence of uneven distribution on the calculation of essential oil contents on the fabrics.

### 3.9. Washing Resistance

A poor washing resistance is one of the shortcomings of fabric modification after finishing, so it is of great significance to improve the washing resistance of finished fabrics. The results of previous studies showed that [19,20] the fragrant fabrics were resistant to washing for approximately 5 to 40 times.

Figure 9 shows the content and loss rate of the essential oils on the fabrics after different washing cycles. As the number of washing cycles increased, the contents of the essential oils on the fabrics gradually decreased. After five cycles, the essential oils lost about 40%, and the loss rate slowed down. After 30 cycles, there were still about 20% of essential oils left on the fabrics, indicating that the fabrics can resist more than 30 washing cycles. 

The nice washing resistance of the fabrics finished with the double-layered microcapsules was due to the smaller particle sizes of the microcapsules, which not only adhered to the surfaces of the fabrics but penetrated deeper into the voids between the fibers. Moreover, the wall materials of chitosan and sodium alginate were a sticky nature; therefore, the microcapsules were not easy to wash off, Meanwhile, during the finishing process, the appropriate curing temperature can make the finishing liquid form a stable and continuous film on the surface of the fabric, covering the essential oil microcapsules, which made the microcapsules more difficult to fall off during the washing process.

### 3.10. Friction Resistance

When talking about fabric finishing, the friction resistance is also important to be evaluated. As can be seen from Figure 10, as the friction cycles increased, the contents and loss rates of the essential oils in the fabric gradually decreased; however, the amounts lost were small. After 100 friction cycles, only about 30% of the essential oils were lost in the fabrics. The ideal friction resistance of the fabrics finished by the double-layered microcapsules was also due to the smaller particle sizes of the microcapsules, as well as the stable and continuous films formed during the finishing process.

### 3.11. The Release of Essential Oils on the Finished Fabrics

It can be seen from Table 2 that the new treated fabrics had strong and suitable fragrances and can still maintain the strong fragrances after one week. After two weeks, the essential oils on the fabrics had a weak scent. After two months of natural storage, the scents could still be smelled, but the intensity of the scents was very low. Six months later, the scents were hardly smelled on the fabrics. Due to the different sensitivity of each person’s smell, it is difficult to quantify the subjective measurements of fragrances. Therefore, a more accurate objective instrument was used to evaluate and analyze the contents of essential oils on the fabrics.

Figure 11 shows the changes and loss of the essential oil contents on the fabrics under natural storage conditions. With increasing time, the contents of essential oils on the fabrics show a downward trend. Within one week after finishing, the essential oils on the fabrics released fast and lost nearly 30%. After one week, the loss rates of essential oils on the fabrics were relatively stable, and only 10% of the essential oils were lost in the second week. From the second week to the fourth week, 20% of the essential oils were lost. After two months, there were still about 30% of the essential oils left on the fabric. When the fabric was placed for half a year, about 10% of the essential oils remained. This result is basically consistent with the subjective evaluation. This is the result of the combined action of the slow-release properties of the double-layer microcapsules and the firm combination of the microcapsules and the fabrics. Fabrics finished with essential oil microcapsules can be naturally placed for two months and still retain their fragrances, with a slow release and long-lasting effects.

## 4. Conclusions

To sum up, a kind of double-layered microcapsule was successfully prepared by using β-cyclodextrin, chitosan and sodium alginate as the composite wall material and lavender essential oil as the core material. The optimal encapsulating conditions providing high encapsulation efficiency were attained using the response surface methodology. The microcapsules prepared in this study had smaller particle size distributions and better thermal stability. Therefore, the double-layered microcapsules could continuously release the active ingredients of the essential oils under normal temperatures and high temperatures for a long time. Furthermore, the cotton fabrics finished by the microcapsules showed excellent washing resistance and rubbing fastness. They can still maintain light fragrances for two months of storage naturally. The double-layered microcapsules prepared in this study can be applied in the fields of textile, food science, biomedicine, healthcare, etc.

## Figures and Tables

**Figure 1 polymers-12-01651-f001:**
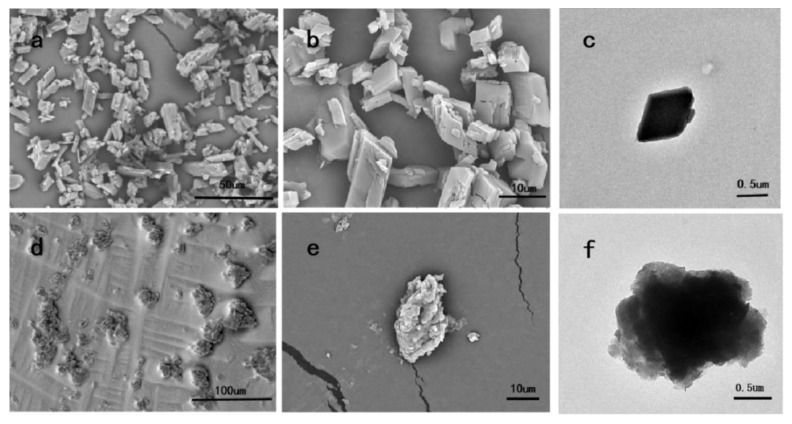
SEM and TEM images of single-layered microcapsules (**a**–**c**) and double-layered microcapsules (**d**–**f**).

**Figure 2 polymers-12-01651-f002:**
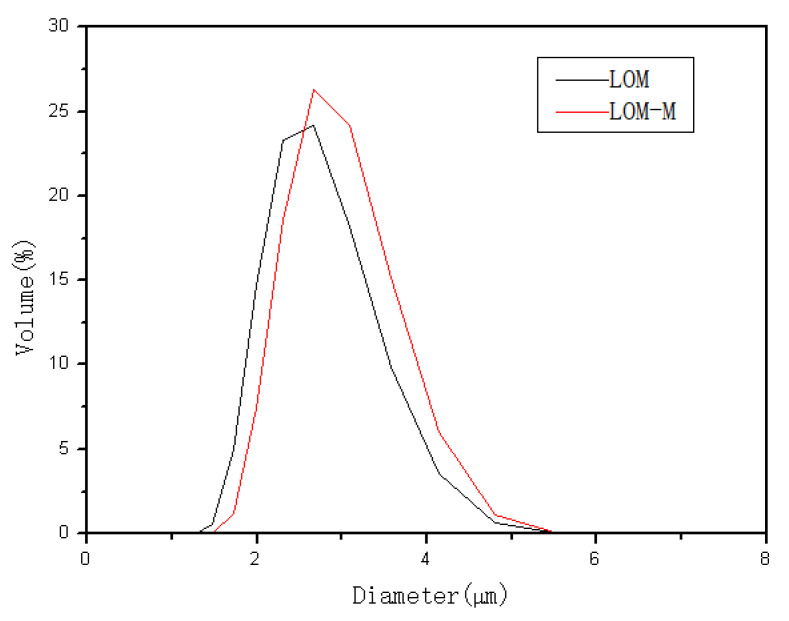
Particle size distribution of single-layered microcapsules (lavender essential oil (LOM) and double-layered microcapsules (LOM-M).

**Figure 3 polymers-12-01651-f003:**
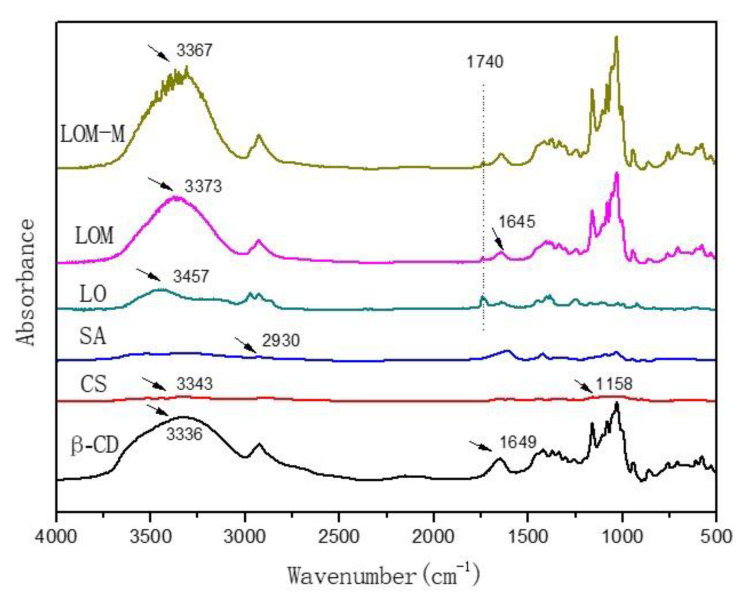
Fourier-transform infrared (FT-IR) spectroscopy of β-cyclodextrin (β-CD), chitosan (CS), sodium alginate (SA), lavender essential oil (LO), single-layered microcapsules (LOM) and double-layered microcapsules (LOM-M).

**Figure 4 polymers-12-01651-f004:**
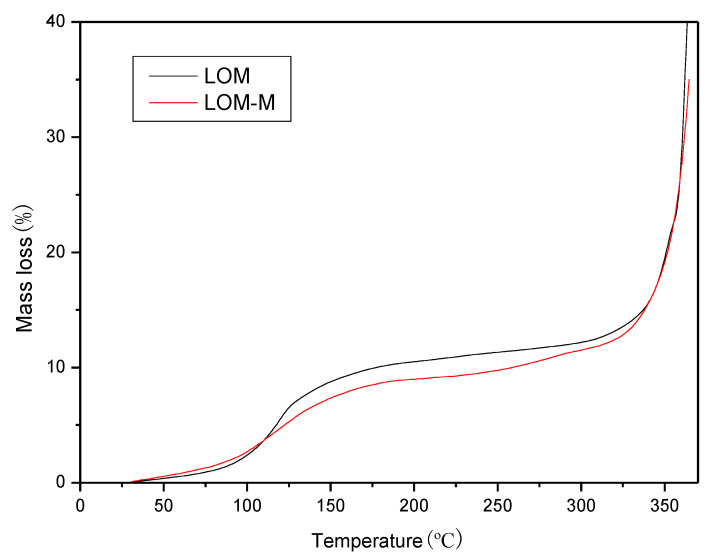
Thermogravimetric (TG) curves of lavender essential oil single-layered microcapsules (LOM) and double-layered microcapsules (LOM-M).

**Figure 5 polymers-12-01651-f005:**
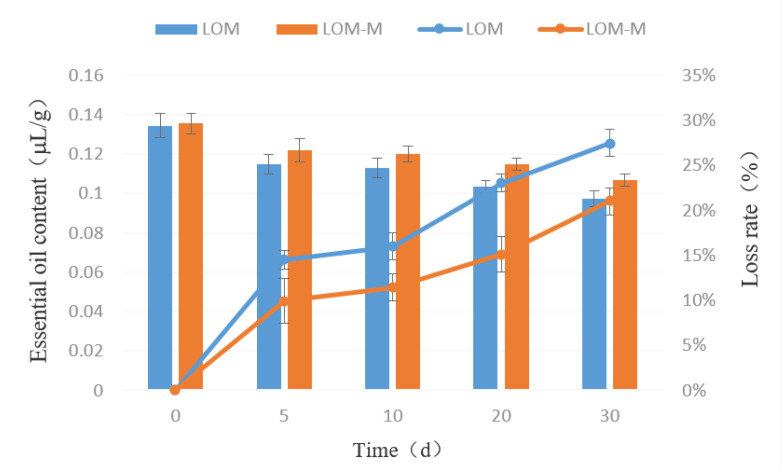
The release of lavender essential oil in single-layered microcapsules (LOM) and double-layered microcapsules (LOM-M) at room temperature.

**Figure 6 polymers-12-01651-f006:**
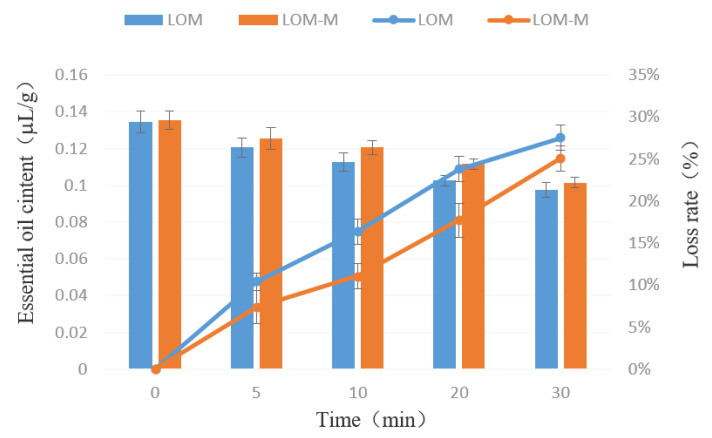
The release of lavender essential oil in single-layered microcapsules (LOM) and double-layered microcapsules (LOM-M) at high temperatures.

**Figure 7 polymers-12-01651-f007:**
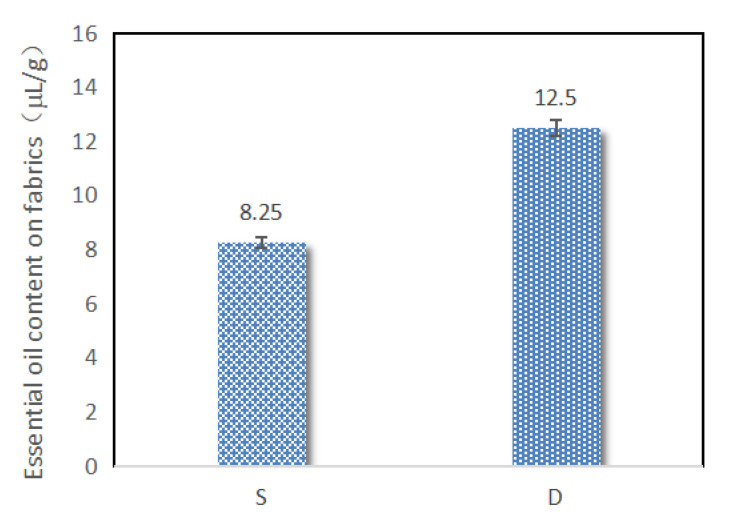
Essential oil contents of fabrics finishing with single-layered (S) and double-layered (D) microcapsules.

**Figure 8 polymers-12-01651-f008:**
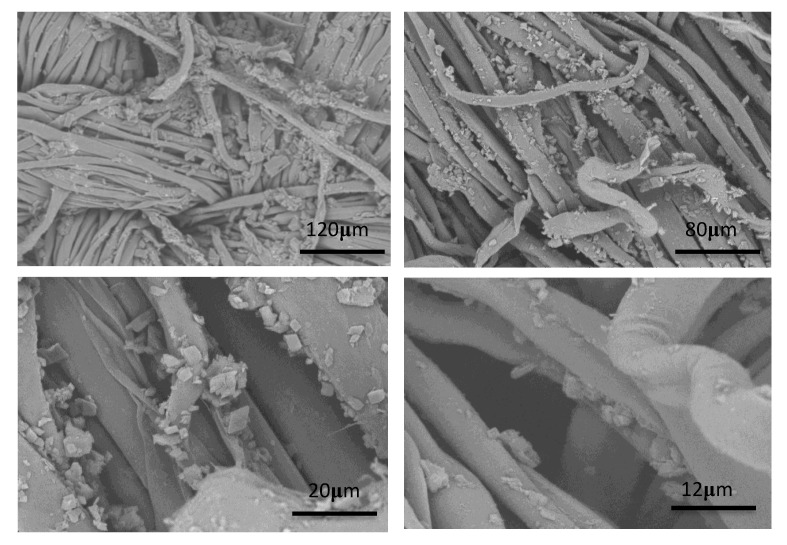
SEM photographs of fabrics finished with double-layered microcapsules.

**Figure 9 polymers-12-01651-f009:**
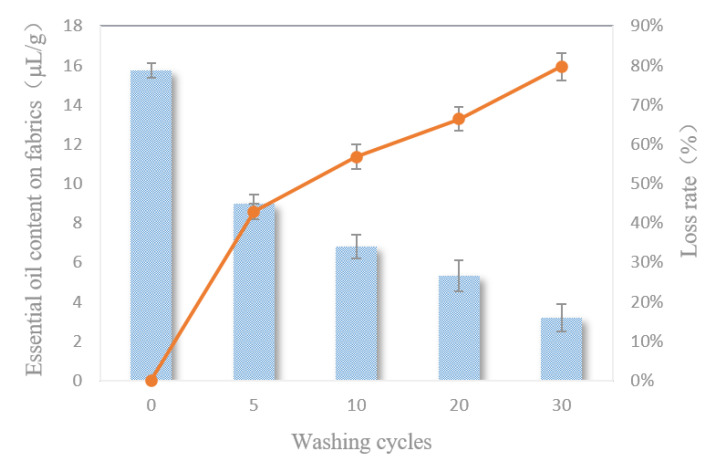
The washing resistance of the finished fabrics.

**Figure 10 polymers-12-01651-f010:**
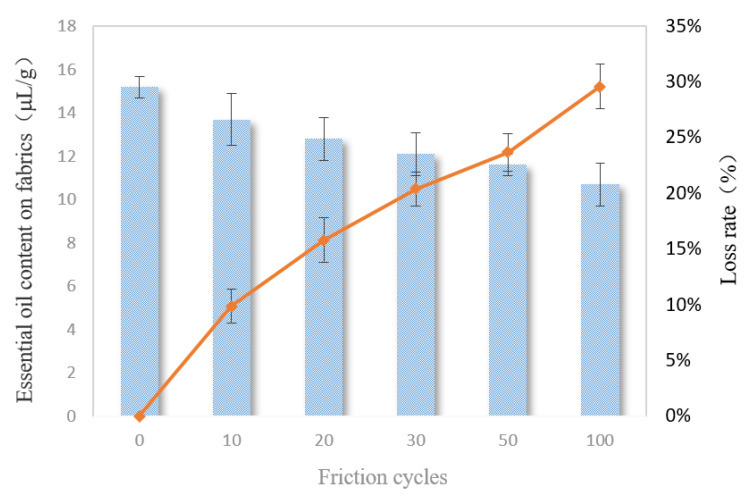
The friction resistance of the finished fabrics.

**Figure 11 polymers-12-01651-f011:**
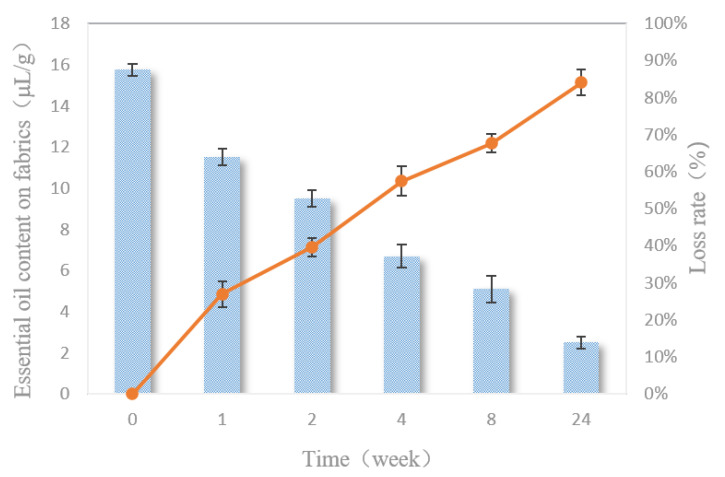
The release of essential oils on the finished fabrics.

**Table 1 polymers-12-01651-t001:** Independent variables and their levels for the Box-Behnken design (BBD) used in the response surface methodology (RSM). LO: lavender essential oil.

	Factors	A: LO Mass (mL)	B: Encapsulating Time (h)	C: Encapsulating Temperature (°C)
Levels				
−1		1	2	40
0		2	3	50
1		3	4	60

**Table 2 polymers-12-01651-t002:** Subjective evaluations of the effects of natural fragrance retention on the finished fabrics.

Storage Time	New Treatment	1 Week	2 Weeks	4 Weeks	8 Weeks	24 Weeks
Evaluation	****	****	***	**	**	*

(Note: ***** strong scent, **** medium scent, *** weak scent, ** very weak scent and * no scent).

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
