# Peer review of "Double-Layered Microcapsules Significantly Improve the Long-Term Effectiveness of Essential Oil"

_polymers, 2020, doi:10.3390/polym12081651_

Round 1

Reviewer 1 Report

The paper is a study on encapsulation of essential oil into double-layered microcapsules of β- cyclodextrin, chitosan and sodium alginate with the aim is to improve its long-term effectiveness. The authors used lavender essential oil as a model system and successfully incorporated it into developed double-layered microcapsules. The morphology, particle size, encapsulation efficiency, thermal stability and sustained release performance of the double-layered microcapsules have been characterized as well. The authors also demonstrated that the obtained microcapsules can be incorporated into cotton fabric materials which showed excellent washing resistance and maintain fragrance for two months. In general, the paper is well structured and written in good scientific style.

I have some minor question concerning the used methodology:

L 97 – provide the amount of essential oil used in the experiment;

L 102 – provide the volume of 1% acetic acid solution;

L 103-104 – “single-layered microcapsule solution” – how this solution was prepared from dried single-layered microcapsules? What were its concentration and volume?

L 105-106 – “was centrifugated, filtrated and washed” – provide rpm and time for centrifugation, type of filtration system used, and details for used solvent for washing;

L 106-108 – describe this step in more detailed and quantitative manner;

L 109-116 – this optimization was performed for single-layered or double-layered microcapsules? This is important especially for the definition of independent variables “time” and “temperature”.

Author Response

1. L97 – provide the amount of essential oil used in the experiment;

   The amount of essential oil was provided, which has been highlighted in yellow in the text.

2. L102 – provide the volume of 1% acetic acid solution;

   The volume of acetic acid solution is 200 mL, which has been highlighted in yellow in the text.

3. L103-104 – “single-layered microcapsule solution” – how this solution was prepared from dried single-layered microcapsules? What were its concentration and volume?

   The relative data have been highlighted in yellow in the text.

4. L105-106 – “was centrifugated, filtrated and washed” – provide rpm and time for centrifugation, type of filtration system used, and details for used solvent for washing;

   The mixture was centrifugated at 9000 rpm for 10 minutes, vacuum filtered with Brinell funnel by a vaccum pump and washed with deionized water, which has been highlighted in yellow in the text.

5. L106-108 – describe this step in more detailed and quantitative manner;

The changes have been highlighted in yellow in the text.

6. L109-116 – this optimization was performed for single-layered or double-layered microcapsules? This is important especially for the definition of independent variables “time” and “temperature”.

   Response surface methodology (RSM) was used to determine the optimal encapsulating conditions of single/double-layered microcapsules, which has been added and highlighted in yellow in the text.

Reviewer 2 Report

The manuscript submitted by the authors is good. They have addressed an important problem such as the microencapsulation of essential oils for different uses. The problem is well exposed and justified in the introduction, indicating in detail the aspects that must be analyzed and studied.The methods and techniques used are well described and have been used correctly for the experiments indicated in the work. The processes have been optimized and the use of microcapsules in cotton threads has been studied to study their capabilities and characteristics.The work is very well presented and written correctly, no errors were found, the style used is good.The conclusions are well described, present good results and justify and confirm that the method used in the work was well planned.

Author Response

The manuscript submitted by the authors is good. They have addressed an important problem such as the microencapsulation of essential oils for different uses. The problem is well exposed and justified in the introduction, indicating in detail the aspects that must be analyzed and studied.The methods and techniques used are well described and have been used correctly for the experiments indicated in the work. The processes have been optimized and the use of microcapsules in cotton threads has been studied to study their capabilities and characteristics.The work is very well presented and written correctly, no errors were found, the style used is good.The conclusions are well described, present good results and justify and confirm that the method used in the work was well planned.

Reviewer 3 Report

Lovely & blinding! My sincere congrats to the respected Authors. Bravo! What a clever (wise) choice of EOs related research topic!

Based on its scientific merit, I can most kindly recommend Your highly informed (knowledgeable) manuscript (MS) for the publishing in a forthcoming issue of Polymers, a truly esteemed journal feat by MDPI. At least in my humble opinion, the priority issue might be seriously taken into account, when it comes to the launching of this particular submission.

Overall the MS is favourably rated: 9 out of 10

Recommendation: Minor Revision (Minor Changes)

To be completely honest with You, the English language itself requires polishing at a moderate extent. In other words, there is yet a room for the language and style improvement. Please, genially address this particular issue as much as You can. 

In addition to this, the Authors are affably requested to consider the citing of the following references throughout the text of their enlightening MS, e.g. within the section 1. Introduction and/or elsewhere: 

 - Cryptogamie Bryologie 2011, 32, 113–117                                               - Industrial Crops and Products 2013, 49, 561–567                                     - Meat Science 2014, 96(3), 1355–1360

If English polished, this MS has a real potential to be clearly recognised and truly appreciated by the members of the global academic/research community. Hopefully, it will earn a number of hetero-citations (= will be frequently cited), once when launched.                                                       

Taken all together, I strongly encourage the respected Authors to submit the revised form of their knowledgeable (highly informed) MS in quite a short time.

Last but not least, very best of (research) luck ahead to You all.

Author Response

  1. To be completely honest with You, the English language itself requires polishing at a moderate extent. In other words, there is yet a room for the language and style improvement. Please, genially address this particular issue as much as You can.

The language has been checked and the changes have been highlighted in yellow(L12-15, L21-23, L29, L49).

  1. In addition to this, the Authors are affably requested to consider the citing of the following references throughout the text of their enlightening MS, e.g. within the section 1. Introduction and/or elsewhere:

The three articles were cited in our manuscript, which has been highlighted in yellow(L30, L32).
